# HEP-JEPA: A FOUNDATION MODEL FOR COLLIDER PHYSICS

**Jai Bardhan, Radhikesh Agarwal,**\* **Abhiram Potula,**\* **Cyrin Neeraj, Subhadip Mitra**
Center for Computational Natural Sciences and Bioinformatics
International Institute of Information Technology, Hyderabad

## ABSTRACT

We present a transformer architecture-based foundation model for tasks at high-energy particle colliders such as the Large Hadron Collider. We train the model to classify jets using a self-supervised strategy inspired by the Joint Embedding Predictive Architecture Assran et al. (2023). We use the JetClass dataset Qu et al. (2022b) containing 100M jets of various known particles to pre-train the model with a data-centric approach — the model uses a fraction of the jet constituents as the context to predict the embeddings of the unseen target constituents. Our pre-trained model fares well with other datasets for standard classification benchmark tasks. We test our model on two additional downstream tasks: top tagging and differentiating light-quark jets from gluon jets. We also evaluate our model with task-specific metrics and baselines and compare it with state-of-the-art models in high-energy physics. Therefore, this work contributes to the development of scientific foundation models by demonstrating how self-supervised transformer architectures can extract deep insights from high-energy physics data.

## 1 INTRODUCTION

Modern collider experiments, such as the Large Hadron Collider (LHC), rely on deep learning to enhance key tasks like jet tagging, track reconstruction, and detector–simulation matching. Traditionally, the high-energy physics (HEP) community has built dedicated deep-learning models using well-curated, labelled simulated data. However, recent advances in large world models (Team et al., 2023; Reid et al., 2024; Garrido et al., 2024) suggest that a unified model capturing 'common knowledge' could improve performance on individual tasks.

Large Language Models (LLMs) like GPT Brown et al. (2020) and BERT Devlin et al. (2018) have demonstrated success in learning generalized language representations through extensive pre-training. Analogously, a foundation model (FM) in HEP could encode the 'language' of particle interactions, detector responses, and physical laws, serving as a versatile tool for experimental data analysis. As the LHC collects ever-increasing amounts of data to probe new or rare processes, training models from scratch becomes computationally expensive. Pre-trained FMs that require only minimal fine-tuning can save significant compute resources and benefit researchers with limited capabilities.

Recent attempts to build FMs for HEP—such as OmniLearn Mikuni & Nachman (2024) and Particle Transformer Qu et al. (2022a)—rely on supervised training with first-principle simulations. However, dependence on labelled data limits their applicability to real experimental data due to inherent simulation deficiencies. Self-supervised learning (SSL) thus offers a more data-driven strategy. For example, Masked Particle Modelling (MPM) Golling et al. (2024) and OmniJet-$\alpha$ Birk et al. (2024) adapt masked language modelling and generative pre-trained transformers for HEP, while contrastive methods akin to the SimCLR framework Dillon et al. (2022a;b; 2024) have also been explored. Nevertheless, SSL approaches face challenges: contrastive methods require careful selection of negative samples to avoid representational collapse, and masked modelling may overemphasize local features. Furthermore, many SSL techniques rely on decoders for input reconstruction, which can divert computational resources toward learning redundant details.

---

\*Radhikesh Agarwal and Abhiram Potula contributed equally to this work.

We train a FM for collider tasks using the Joint Embedding Predictive Architecture (JEPA) paradigm. Originally proposed in Ref. Assran et al. (2023) for image perception, JEPA learns predictive representations by modelling missing embeddings directly in the latent space—without a decoder or full input reconstruction. This design enhances computational efficiency and yields more abstract, transferable features, as demonstrated in images, videos Bardes et al. (2024), and point clouds Saito & Poovvancheri (2024).

Furthermore, our work serves as a compelling example of a scientific foundation model tailored for agentic AI. By learning high-level, transferable representations of particle interactions in a self-supervised manner, it paves the way for AI systems that can autonomously generate hypotheses and drive scientific discovery—key goals in advancing agentic AI for research.

Our contributions in this paper can be summarised as

1. We adapt JEPA for HEP collider tasks and introduce a foundation model called HEP-JEPA. Most collider tasks require analysing scores of jets (highly challenging but necessary for physics experiments) produced in high-energy particle collisions. We use the JetClass dataset Qu et al. (2022b) to pre-train our foundation model by providing parts of each jet sample as the context for the model to predict the rest of the jet correctly.

2. We present a framework to test model choices and HEP-JEPA performance with detailed ablations and comparisons with different training paradigms on few-shot learning tasks.

3. We evaluate HEP-JEPA on two important downstream tasks: top tagging and differentiating light-quark jets from gluon jets. Top tagging is necessary for practically all new physics searches, and achieving a good discriminator of light jets is one of the significant challenges in the domain. We evaluate the model performance over reference datasets and task-specific metrics.

## 2 JEPA PRE-TRAINING PARADIGM

JEPA learns to predict the embedding of a signal $y$ from a compatible signal $x$, using a predictor network conditioned on additional variables $z$ to facilitate prediction. Instead of predicting $y$ directly, JEPA predicts in representation space, which enables it to learn abstract meaningful representations of inputs, making it ideal for some downstream tasks. Our HEP-JEPA is one instantiation of the paradigm to work with particle jets by masking a fraction of jet constituents. Similar to the joint embedding architecture, JEPA is susceptible to representation collapse, which we bypass using an exponential-moving-average teacher design for the $y$ encoder.

## 3 HEP-JEPA MODEL

Figure 1 shows the complete HEP-JEPA framework, which works as follows. For every input set (jet) of particles (each with 2 coordinates $\eta, \phi$ — pseudorapidity and azimuthal angle — and a set of kinematic variables), we first divide it into geometrical patches based on the coordinates of each particle. Each patch of the jet is encoded using a small permutation invariant model to form patch tokens. These tokens are then divided into non-overlapping context ($x$) and target ($y$) sets and encoded by their respective encoders. A predictor is used to predict the embedding of the target tokens. The encoders are implemented using a transformer encoder architecture.

### 3.1 PARTICLE GROUP TOKENISER

Let us consider a jet $J$ consisting of $n$ particles, where each particle is represented by a vector $p_i \in \mathbb{R}^7$:

$$p_i = (\eta_i, \phi_i, m_i, \ln p_{T_i}, \ln E_i, \ln \frac{p_{T_i}}{p_{T_J}}, \ln \frac{E_i}{E_J}, \Delta_{R_{iJ}}),$$

where $p_T$ denotes the momentum component transverse to the beam direction, $m$ denotes the mass, $E$ is the energy, and $\Delta_{R_{iJ}}$ is the distance between the particle and the jet axis in the $(\eta, \phi)$ plane. We introduce a patchification and tokenisation strategy to capture meaningful *patch-level* particle

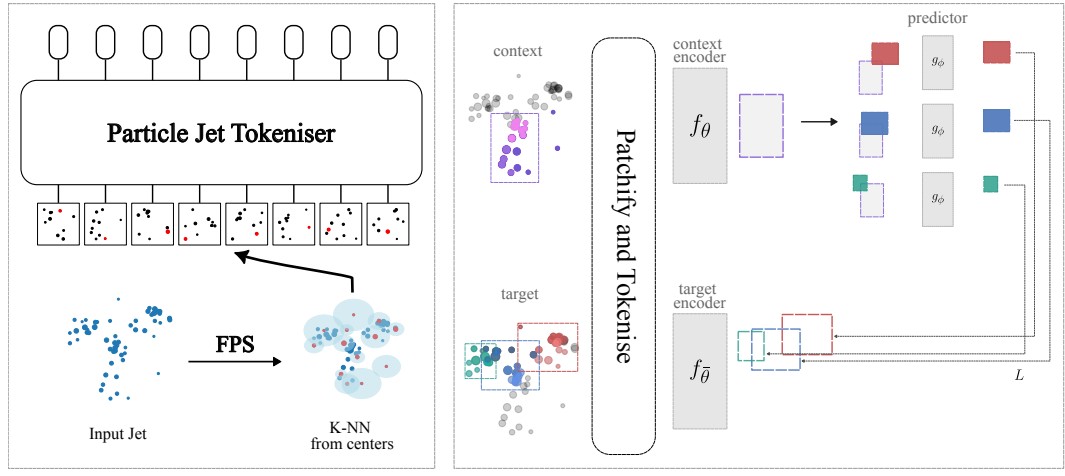

**Figure 1:** Schematic diagram illustrating the working of the HEP-JEPA model. The model has a structure similar to vision transformers. In the first step, the entire jet is divided into patches using a particle jet tokeniser. These tokens are then masked to form the context and target blocks. Each block is fed into the respective encoder to generate the embeddings. The context embedding, along with the special mask tokens, is used by the predictor to predict the embedding of the masked target blocks.

interactions. The tokeniser $T : \mathbb{R}^{n \times 7} \to \mathbb{R}^{c \times d}$ maps the raw particle features to $c$ token embeddings of dimension $d$.

First, we employ Farthest Point Sampling (FPS) to select $c$ centre particles that maximise the jet's phase space coverage. We construct a local group of $k$ particles for each centre using the $k$-nearest neighbours in the $(\eta, \phi)$ plane. These groups are then normalised by subtracting their respective centre coordinates:

$$p_i' = p_i - p_k, \quad \forall i \in G_k$$

where $G_k$ denotes the group of particles associated with centre $k$. To obtain permutation-invariant token embeddings, we process each normalised group through a small PointNet encoder, $E$, consisting of shared multilayer perceptions (MLPs) followed by max-pooling operations:

$$t_k = \max \left( \{ \text{MLP}(p_i') \mid p_i' \in G_k \} \right).$$

In our case, $c$ is not a fixed number as it varies with the size of the particle jet. We sample only a small fraction of the points as the possible centres for the jet.

## 3.2 JEPA FRAMEWORK

### 3.2.1 TOKEN CONSTRUCTION

**Target construction:** Our masking strategy operates on the token sequence $T = \{t_1, \ldots, t_k\}$ to create complementary context and target regions. The $c$ groups of the jet $J$ is fed through the target encoder $f_{\bar{\theta}}$ to obtain a corresponding patch-level embedding $\boldsymbol{s}_y = \{\boldsymbol{s}_{y_1}, \ldots, \boldsymbol{s}_{y_c}\}$, where $\boldsymbol{s}_{y_k}$ is the representation associated with the $c^{\text{th}}$ group. For each jet, we sample $M$ (possibly overlapping) blocks from the target representations $\boldsymbol{s}_y$ with random scale $s \in [0.15, 0.2]$ and random aspect ratio $r \in [0.75, 1.5]$. As has been shown previously, it is essential to mask the output of the target encoder and not the input.

**Context construction:** The context block is sampled with a random scale $s \in [0.4, 0.75]$ from the set of all tokens. Since the target and the context are selected independently, they may overlap significantly. We remove overlapping regions between the source and target tokens to prevent trivial learning. These tokens are fed to the context encoder, $f_{\theta}$, to obtain the corresponding representations $\boldsymbol{s}_x = \{s_{x_j}\}_{j \in B_x}$, where $B_x$ is the mask associated with block $x$. The context encoders are constructed as transformer models .

Following Point-JEPA (Saito & Poovvancheri, 2024), we also utilise a greedy sequencing algorithm to ensure spatial coherence when performing the contiguous masking of source and target tokens. Below we summarise the main steps of the sequencer.

1. Initialise with token $t_i$ minimizing $\sum_i \text{coord}(t_i)$.
2. Iteratively select the next token based on the minimum distance on the $(\eta, \phi)$ plane.
3. Maintain disjoint sets between context and target tokens.

### 3.2.2 ENCODER ARCHITECTURES

Our framework consists of three primary components operating in representation space:

**Context and target encoders:** Our context and target encoders are transformer models for encoding the tokens in a jet sample. We use eight registers with each transformer encoder, following Ref. Darcet et al. (2024). We also introduce a physics bias matrix from Ref. Qu et al. (2022a); however, our implementation differs since we calculate pairwise interactions between tokens (groups) instead of particles. This requires calculating the four-momentum vectors of the entire group. The bias terms are:

$$\Delta_{R_{ij}} = \sqrt{(\eta_i - \eta_j)^2 + (\phi_i - \phi_j)^2},$$
$$k_T = \min(p_{T_i}, p_{T_j})\Delta_{R_{ij}},$$
$$z = \min(p_{T_i}, p_{T_j})/(p_{T_i} + p_{T_j}),$$
$$m^2 = (E_i + E_j)^2 - \|\mathbf{p}_i + \mathbf{p}_j\|^2,$$

where $\mathbf{p}_i$ denotes the momentum of the $i^{\text{th}}$ particle and $\Delta_{R_{ij}}$ is the distance between the $i^{\text{th}}$ and $j^{\text{th}}$ particles in the $(\eta, \phi)$ plane. The bias for the added registers is set to 0.

**Predictor:** The predictor predicts the representations of targets with the help of context blocks. For a given target block $s_y(i)$ corresponding to a target mask $B_i$, the predictor $g_\phi(\cdot, \cdot)$ takes the context encoder $s_x$ and a mask token for each patch we wish to predict $\{m_j\}_{j \in B_i}$ as inputs, and outputs a patch-level prediction $\hat{s}_y(i) = \{\hat{s}_{y_j}\}_{j \in B_i} = g(s_x, \{m_j\}_{j \in B_i})$. The mask tokens are parametrised by a shared learnable vector with an added positional embedding. Since we want to predict for each of the $M$ blocks, we apply the predictor $M$ times.

**Loss function:** The learning objective is formulated entirely in the embedding space as

$$L = \frac{1}{M} \sum_{i=1}^{M} D(\hat{s}_y^{(i)}, s_y^{(i)}),$$

where $D$ is the smooth L1 loss between predicted embeddings $\hat{s}_y^{(i)} = g(s_x, \{m_j\}_{j \in B_i})$ and target embeddings $s_y^{(i)} = \{s_{y_j}\}_{j \in B_i}$. By operating in the representation space rather than the particle space, we focus the learning on physically relevant features while avoiding the computational overhead of full reconstruction.

## 4 PRE-TRAINING HEP-JEPA

We pre-train our models on the JetClass dataset Qu et al. (2022b). JetClass is an extensive collection of jets clustered from simulated proton-proton collisions at the LHC. It contains 100M training samples divided into 10 jet classes, covering light jets (from gluons and light quarks) and heavy-particle jets from the top quark or the Higgs, $W$, and $Z$ bosons. Each jet is reconstructed using the anti-$k_T$ algorithm Cacciari et al. (2008) (with jet radius $R = 0.8$) after incorporating detector effects from DELPHES de Favereau et al. (2014), the detector simulator. The dataset provides detailed per-particle features, including kinematics (energy-momentum four-vectors), particle identification (5-class encoding), and trajectory displacement parameters for tagging heavy-flavor jets. It is split into training (100M jets), validation (5M), and test (20M) sets.

**The model and its training:** For the model, we take a standard transformer architecture of 12 transformer blocks with 2.5M parameters in total. We train the model for 4 epochs with an effective batch size of 2048, corresponding to roughly 200k training steps. We take a cosine decay scheduler with a linear warm-up for 15k steps. Our training was complete in 320 GPU hours on RTX 2080Ti.

# 5 EXPERIMENTS

## 5.1 EVALUATIONS ON JETCLASS DATASET

### 5.1.1 FEW-SHOT LEARNING

We perform a few-shot evaluations on the JetClass dataset. As the baseline model, we take our architecture trained from scratch in a supervised fashion to eliminate the effects of other external factors. In both these models (ours and the baseline), we attach a classification head to the student backbone constructed as two class attention blocks followed by a MLP layer. We consider two regimes to perform the few shot evaluation:

(1) *frozen:* where the pre-trained backbone is not updated – only the classification head is trained, and

(2) *fine-tuned:* where the pre-trained backbone is simultaneously updated with the classification head.

We conduct the experiment at different label fractions of the JetClass dataset – $0.05\%$ $0.5\%$, $2\%, 10\%$, and $100\%$.

Table 1: JetClass Metrics: Peak validation accuracies attained by the benchmark models on the JetClass dataset. We run experiments for different levels of few-shot learning. We provide different fractions of labels from the JetClass dataset to train the model from scratch for the classification task. In each experiment, a pre-trained HEP-JEPA model is fine-tuned on the same fraction of labels.

| % OF LABELS | MODEL | ACCURACY |
|---|---|---|
| 0.05% (5K) | FROM SCRATCH | 0.505 |
| | HEP-JEPA, FINE-TUNING | 0.564 |
| 0.5% (50K) | FROM SCRATCH | 0.586 |
| | HEP-JEPA, FINE-TUNING | 0.624 |
| 2% (2M) | FROM SCRATCH | 0.668 |
| | HEP-JEPA, FINE-TUNING | 0.669 |
| 10% (10M) | FROM SCRATCH | 0.683 |
| | HEP-JEPA, FINE-TUNING | 0.685 |
| 100% (100M) | FROM SCRATCH | 0.698 |
| | HEP-JEPA, FINE-TUNING | 0.698 |

Table 1 shows the macro accuracy obtained by the methods. The fine-tuned HEP-JEPA model consistently outperforms the model trained from scratch. The difference is significant in few-shot learning tasks (i.e., $0.05\%$- and $0.5\%$-label cases), where HEP-JEPA shows $4 - 6\%$ better accuracies than the model trained from scratch. To illustrate this, we show Figure 2, where we plot the validation loss against the training step for the two benchmark models training in a few-shot learning setting for jet classification on the JetClass dataset with $0.5\%$ labels. However, as the fraction of labels for training increases, the performance difference between the benchmarks reduces. HEP-JEPA performs almost identically to the model trained from scratch when the complete set of labels is available for training.

## 5.2 TRANSFER LEARNING EVALUATIONS ON DOWNSTREAM TASKS

We also test the model's capabilities for generalising and its performance on different datasets.

Table 2: *Results for downstream tasks*: Peak validation accuracies for the benchmark models on the TQTR dataset Kasieczka et al. (2019b) for top tagging task (left panel) and a reference quark-gluon tagging dataset Komiske et al. (2019) (right panel) using 100% of the data samples. "FROZEN" after a model name indicates that the pre-trained was not updated while training, whereas the tag "FINE-TUNED" indicates that both the classifier head and the pre-trained backbone were updated during the training. We also report the performances of two state-of-the-art models on top tagging from Refs. Qu & Gouskos (2020); Qu et al. (2022a), indicated with (*), for a comparison to the HEP-JEPA performance.

| TOP TAGGING | | QUARK-GLUON TAGGING | |
|---|---|---|---|
| MODEL | ACCURACY | MODEL | ACCURACY |
| FROM SCRATCH | 0.927 | FROM SCRATCH | 0.819 |
| SUPERVISED, FROZEN | 0.928 | SUPERVISED, FROZEN | 0.823 |
| SUPERVISED, FINE-TUNED | 0.938 | | |
| HEP-JEPA, FROZEN | 0.928 | HEP-JEPA, FROZEN | 0.821 |
| HEP-JEPA, FINE-TUNED | 0.929 | | |
| PARTICLENET (*) | 0.940 | PARTICLENET (*) | 0.840 |
| PART (*) | 0.944 | PART (*) | 0.843 |

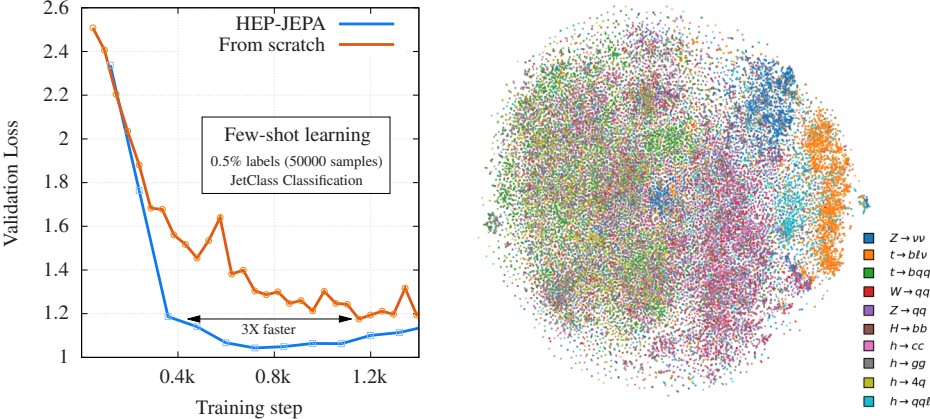

Figure 2: *Left panel*: Validation loss vs. training step for the two benchmark models training in a few-shot learning setting for jet classification on the JetClass dataset with 0.5% labels (i.e., 50000 training samples). The validation loss falls quickly for the HEP-JEPA model — it achieves the same minimum validation loss as the model trained from scratch three times faster. *Right Panel*: t-SNE plot of the pooled embedding obtained for samples within the JetClass dataset.

### 5.2.1 TOP TAGGING

The top quark is the heaviest known particle of the Standard Model. Once produced, it quickly decays inside the collider and predominantly produces a three-pronged jet (from its hadronic, i.e., $t \rightarrow Wb \rightarrow qq'b$ (three-body) decays). The dominant background processes also often produce similar jet signatures, making it challenging to achieve high signal-to-background ratios. To evaluate how HEP-JEPA performs in tagging top jets, we use the Top Quark Tagging Reference (TQTR) dataset Kasieczka et al. (2019b), which consists of 2M samples of jets originating from hadronic decays of the top quark as well as ones from lighter quarks and gluons, with the recommended split of 1.2M samples for training, 400k each for validation and testing.

We evaluate the performance of our model architecture with two benchmark models: one trained on the JetClass dataset in a supervised fashion and the other trained on the TQTR dataset from scratch. We also compare how fine-tuning affects the performance of these benchmark models.

Left panel of Table 2 shows the top-tagging metrics obtained using $100\%$ of the dataset. The fine-tuned versions of the benchmark models perform better than their frozen versions. The fine-tuned HEP-JEPA attains a better accuracy score than both the supervised model trained from scratch and the frozen HEP-JEPA. However, we see that the fine-tuned supervised model shows better performance than the other benchmarks. To understand this, we should keep in mind that 1) the kinematic range of jet samples from the TQTR dataset (transverse momenta $p_T \in [550, 650]$) is smaller than that of the JetClass dataset ($p_T \in [500, 1000]$) and 2) HEP-JEPA is trained on the auxiliary task predicting missing parts of the jet samples, whereas the supervised model is trained for the particular classification task. Both factors could give a slight edge to the fine-tuned supervised benchmark model. We also show the accuracy scores from the domain state-of-the-art models Qu & Gouskos (2020); Qu et al. (2022a) for reference.

### 5.2.2 QUARK VS. GLUON JET TAGGING

Accurately tagging light jets is one of the most important open problems in collider physics — it is not yet possible to make out (at least reliably) whether a light jet originated in a light quark ($u$, $d$, or $s$) or a gluon at the LHC. The ability to do so would open a new window to new physics searches and significantly enhance the sensitivity of rare process searches. To evaluate how HEP-JEPA perform on this issue, we use the quark-gluon tagging dataset Komiske et al. (2019) containing 2M samples of quark and gluon jets modelled using PYTHIA Sjöstrand et al. (2015) without any detector effects.

As in the case of top tagging, we evaluate the performance of two benchmark models, i.e., a fully supervised model and a model trained using JEPA on the JetClass dataset for this task.

Right panel of Table 2 shows the accuracies of quark-gluon tagging when the models are trained with $100\%$ of the samples. The numbers show a similar trend to the top tagging case. The fine-tuned HEP-JEPA outperforms the model trained from scratch but slightly falls short of when the latter is further fine-tuned on the dataset. We also show the accuracy scores from the domain state-of-the-art models Qu & Gouskos (2020); Qu et al. (2022a) for reference.

### 5.3 VISUALISATION

We visualise the representation learned by HEP-JEPA on 50k samples of JetClass sampled uniformly from each class. We construct the embedding for a sample by concatenating the max and mean pooling of the outputs of the context encoder and apply t-SNE on the pooled embedding. We visualise the results in Right panel of Figure 2. We observe that events that contain lepton(s) are pushed to the right, while hadronic events are more towards the left.

## 6 ABLATION TESTS

We perform ablations on our model choices to understand how they affect the model performance. Our ablation test setup is as follows. For all tests, we pre-train our model for an epoch (roughly 48000 training steps) on the 100M training samples of the JetClass dataset. We estimate the final performance using a SVM linear classifier trained on 50k training samples and evaluate on 50k validation samples obtained by stratified random sampling. Given our resource constraints, we make these reductions to allow us to prototype ablation decisions quickly.

**Test I – Context and Target selection (masking) strategy:**  We test two strategies for target masking — *random* and *contiguous* selection. In *random* selection, we randomly select a fraction of target tokens from all the tokens to mask. In *contiguous* selection, we select a fraction of tokens corresponding to neighbouring regions in the $(\eta, \phi)$ plane. The latter is implemented using the Point Sequencer from Ref. Saito & Poovvancheri (2024). We also run a preliminary investigation on the effect of different context sample ratios.

**Test II – Number of targets:**  We also test the number of target tokens that the model needs to predict for each context token; specifically, we check with 1, 4, and 8 target tokens. The results indicate that the number of target tokens does not significantly influence performance.

Table 3: Ablation tests, Set I: *Top Panel*: Peak validation accuracies for the number of targets and masking strategies. The context sample ratio fixed to the range $[0.15, 0.2]$ and the target sample ratio to $[(0.4, 0.75]$. *Bottom Panel*: Peak validation accuracies for different context sample ratio. The target sample ratio is in the range $(0.15, 0.2)$ with the number of targets set to 1.

| STRATEGY | FREQUENCY | ACCURACY |
|---|---|---|
| RANDOM | 4 | 0.579 |
| CONTIGUOUS | 1 | **0.581** |
| CONTIGUOUS | 4 | 0.557 |
| CONTIGUOUS | 8 | 0.562 |

| STRATEGY | CONTEXT SAMPLE RATIO | ACCURACY |
|---|---|---|
| CONTIGUOUS | $(0.4, 0.75)$ | 0.557 |
| CONTIGUOUS | $(0.85, 1.0)$ | **0.563** |

From these tests, we find that a contiguous target masking strategy with one target is a better model choice. After the target is selected, a sample ratio in the range $[0.85, 1.0]$ performs better.

**Test III – Physics bias for the attention mechanism:** We also analyse the impact of physics bias on the model's performance. Since each token corresponds to a group of particles, we calculate the pairwise bias terms for groups of particles by first summing the four-vectors of the particles within a group to get the group-level four-vector. Left panel in Table 4 show that the model performs $\approx 2\%$ better after including the physics bias.

Table 4: Ablation tests, Set II: Peak validation accuracies for different ablation tests. *Left panel* – with/without physics bias for the attention mechanism. *Mid panel* – with/without registers for the attention mechanism. *Right panel* – with/without integrating augmentations for the data preprocessing

| PHYSICS BIAS | ACCURACY | REGISTERS | ACCURACY | AUGMENTATIONS | ACCURACY |
|---|---|---|---|---|---|
| ✔ | **0.570** | ✔ | **0.576** | ✔ | 0.553 |
| ✗ | 0.557 | ✗ | 0.557 | ✗ | 0.557 |

**Test IV – Integrating registers with our transformer model:** Adding additional tokens to the input sequence of the Vision Transformers has been shown to yield smoother feature maps and attention maps for downstream visual processing Darcet et al. (2024). Given the similarity of our transformers with vision transformers, we investigate the impact of integrating register tokens with our transformer blocks. Middle panel in Table 4 shows that the model performance increases by $\approx 2\%$ with registers.

**Test V - Physics-inspired augmentations:** We also perform a preliminary study of the impact of adding physics-based data augmentations to the jets. Particularly, we test a combination of rotation, smearing, and boosting. However, we do not find any significant improvement in the model's performance by training it on the augmented data.

Right panel in Table 4 shows that this test is inconclusive — the model performance is similar in both cases.

# 7 RELATED WORKS

Foundation models have transformed artificial intelligence by enabling general-purpose learning across diverse tasks and domains Bao et al. (2022); Caron et al. (2021); Kim et al. (2024); Touvron et al. (2023). Various paradigms have emerged, including generative approaches like Masked

Autoencoders (MAE) He et al. (2021) and its 3D extension Point-MAE Pang et al. (2022), multi-scale variants such as Point-M2AE Zhang et al. (2022), and contrastive methods exemplified by SimCLR Chen et al. (2020). Recently, JEPA Assran et al. (2023) was introduced to learn abstract representations while addressing shortcomings of earlier self-supervised models, and it has been adapted to videos (Bardes et al., 2024) and point clouds (Saito & Poovvancheri, 2024).

In fundamental sciences, similar models have been developed in biology Rives et al. (2021); Ross et al. (2022); Bhattacharya et al. (2024), chemistry Liao et al. (2024); Irwin et al. (2022), astronomy Parker et al. (2024), and the modelling of dynamical systems Subramanian et al. (2023); McCabe et al. (2024). In high-energy physics (HEP), early works include Masked Particle Modelling (MPM) Golling et al. (2024); Leigh et al. (2024), which trains on the JetClass dataset by masking a subset of particle features and using a transformer to predict them. Another approach, OmniJet-$\alpha$ Birk et al. (2024), employs an autoregressive transformer to generate tokenised jets similarly to GPT models Brown et al. (2020). Other methods leverage contrastive learning techniques Dillon et al. (2022a;b; 2024); Harris et al. (2024), while OmniLearn Mikuni & Nachman (2024) uses supervised training with first-principle HEP simulations to achieve performance comparable to the Particle Transformer (ParT) Qu et al. (2022a) with faster training. Concurrently, Ref. Katel et al. (2024) adapts JEPA for top tagging by pre-training on 1% of the top and light jet samples from JetClass and evaluating on the TQTR dataset. In contrast to their physics-motivated predictor based on subjets, our approach attains superior top tagging accuracy. We further evaluate our model on quark-gluon tagging and present detailed ablations.

## 8  Conclusions

High-energy physics is a data-intensive and experimentally challenging domain, where a successful foundation model can accelerate the search for new physics and lead to fundamental insights in physics. Particularly in jet physics, foundation models can effectively capture inherent complex patterns and reshape how we approach challenging tasks in collider physics, thereby reducing the use of computational resources significantly. In this paper, we introduced the JEPA framework to the domain of HEP and showed its capabilities with two crucial downstream tasks: top tagging and quark-gluon tagging. We also showed its effectiveness in few-shot learning settings for jet classification tasks.

The JEPA paradigm was tested and thoroughly evaluated on an extensive HEP dataset for the first time. Though its current performance indicates scopes for improvement compared to other state-of-the-art task-specific methods like ParT Qu et al. (2022a) (in the absolute sense), JEPA-based training has the ability to learn better abstract representations and shows superior scalability to real experimental data. These reasons impel us to improve HEP-JEPA further.

JEPA as a paradigm is also largely independent of the underlying model backbone and, therefore, can benefit from improvements to the underlying model architecture. Evaluating HEP-JEPA on other tasks, such as unfolding detector measurements to improve first-principle simulations and anomaly detection using weakly supervised methods, could further validate its generalisability [this can be easily done with publically available datasets Andreassen et al. (2019) and Kasieczka et al. (2019a) respectively]. Additionally, extending HEP-JEPA to generative tasks and event classification is the next step to make it a truly cross-task FM for collider physics.

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
