# OpenReview forum: "HEP-JEPA: A foundation model for collider physics"
_ICLR.cc/2025/Workshop/AgenticAI — ICLR 2025 Workshop AgenticAI Poster_

### Official Review · Reviewer_8ezm · 2025-03-03

**Rating:** 4
**Confidence:** 3

**Review:**

The paper presents HEP-JEPA, a foundation model based on transformer architectures for tasks in high-energy physics, specifically at particle colliders such as the Large Hadron Collider. The model employs a self-supervised learning strategy inspired by the Joint Embedding Predictive Architecture, trained on the JetClass dataset. HEP-JEPA is compared against state-of-the-art models in the field with task-specific metrics and baselines. The paper also includes ablation studies to analyze the impact of different architectural choices and training strategies.

My main concerns about this paper are related to the experimental results, specifically the following points.

1.	Few-Shot Learning Performance. Although the fine-tuned HEP-JEPA model consistently outperforms the model trained from scratch (Table 1), in few-shot learning tasks (0.05%- and 0.5%-label cases), the accuracy is around 0.5, which is close to random results. This raises concerns about whether the experimental results are meaningful for real-world problems where reliable performance is required even with limited labeled data.

2.	Limited Improvement Over Other Methods. The paper compares HEP-JEPA with baseline supervised models and existing physics foundation models. While the fine-tuned HEP-JEPA achieves better accuracy than the supervised model trained from scratch and the frozen HEP-JEPA, the improvement is relatively small. Additionally, state-of-the-art models such as PARTICLENET and PART outperform HEP-JEPA by a significant margin. The authors should provide further analysis to clarify whether the increased complexity of JEPA-based training is effective.

3.	Comparison with More Methods. HEP-JEPA is a self-supervised based method, are there other self-supervised approaches in this field? In addition, the related work section mentions techniques such as Masked Particle Modelling and OmniJet-$\alpha$, but there is no empirical comparison with these methods.

4.	Ablation Study. Some architecture in the ablation study shows only minor improvements or even no improvement (e.g., physics-inspired enhancements).

5.	Computational Scalability. The training process requires significant GPU hours (320 hours on RTX 2080Ti), making it computationally expensive. How does this scale to larger datasets and practical deployment scenarios?

---

### Official Review · Reviewer_ubnq · 2025-03-03
**HEP-JEPA: Good integration of domain knowledge, Lack novel method and significant experimental results**

**Rating:** 6
**Confidence:** 4

**Review:**

Focus on jets classification task, the authors explores how self-supervised learning can learn deep insights from the dataset. The authors propose HEP-JEPA framework to conduct self-supervised pretraining for jets classfication. The framework includes physical insights to enhance the performance. Experimental results show that the pretraining can greatly increase the training efficiency, benefiting to final results.



## Strengths:

* This model integrated several physical insights for framework design.
* The ablation study show that the proposed constraints, ie, physical bias and registers can robust the results.

## Weaknesses:
* The proposed framework is not novel. It is a simple adaptation from existing SSL frameworks.
* How the SSL performs are not compared. The authors use few shot learning to demonstrate the effectiveness of SSL. However, comparing few shot learning with training from scratch cannot directly demonstrate the effectiveness of SSL.
* Compared to SOTAs, the results are not significant.
* Include abbreviation for FPS when it first appears.
* f_{\theta} is used without definition. Please include an overview introduction of encoder.

---

### Official Review · Reviewer_stpt · 2025-03-04
**Review for HEP-JEPA**

**Rating:** 7
**Confidence:** 3

**Review:**

The paper introduces HEP-JEPA, a novel transformer-based foundation model designed for collider physics tasks. It adapts the Joint Embedding Predictive Architecture (JEPA) paradigm to learn high-level representations of particle jets. Pre-trained on the extensive JetClass dataset, the model is evaluated on key downstream tasks such as jet classification, top tagging, and quark–gluon discrimination. By leveraging self-supervised learning and innovative tokenization/masking strategies (including physics-inspired biases), HEP-JEPA demonstrates competitive performance, particularly in few-shot learning settings, and shows promise for more scalable and data-efficient analyses in high-energy physics.

The paper exhibits strong quality through its rigorous methodology, comprehensive experimental setup, and detailed ablation studies that justify various design choices. It shows sufficient clarity with a systematic presentation of the model’s architecture, the pre-training paradigm, and the downstream evaluations. Adapting the JEPA framework to the challenging domain of collider physics is a novel approach, especially given the integration of physics-specific components within a transformer architecture.

### Pros:

- This paper applies the JEPA paradigm from computer vision to high-energy physics in a novel way.

- This paper demonstrates effectiveness through thorough experiments on jet classification, top tagging, and quark-gluon discrimination.

- This work provides extensive ablation studies that offer insights into the effects of different model components and design choices.

- The paper is well-structured, with detailed descriptions of methodology, experimental setups, and results.

### Cons:

- In certain downstream tasks, the absolute performance improvement over specialized supervised models is limited.

- Heavy reliance on the JetClass dataset might restrict direct applicability to real experimental data without further validation.

- The training process, while efficient in few-shot regimes, still requires substantial computational resources (e.g., 320 GPU hours), which could be a barrier for smaller research groups.

---

### Decision · Program_Chairs · 2025-03-05

Accept (Poster)